# Comparison of the Gut Microbiome between Atopic and Healthy Dogs—Preliminary Data

**DOI:** 10.3390/ani12182377

**Published:** 2022-09-12

**Authors:** Ana Rostaher, Yasser Morsy, Claude Favrot, Stefan Unterer, Manuela Schnyder, Michael Scharl, Nina Maria Fischer

**Affiliations:** 1Clinic for Small Animal Internal Medicine, Vetsuisse Faculty, University of Zurich, 8057 Zurich, Switzerland; 2Department of Gastroenterology and Hepatology, University Hospital of Zurich, University of Zurich, 8057 Zurich, Switzerland; 3Institute for Parasitology, Vetsuisse Faculty, University of Zurich, 8057 Zurich, Switzerland

**Keywords:** dog, atopic dermatitis, gut microbiota, bacterial diversity, bacterial composition

## Abstract

**Simple Summary:**

Atopic dermatitis is a common inflammatory and itchy skin disease, constituting a global issue that affects up to 15% of the general human and dog population. The pathogenesis of this disease is known to be multifactorial and not only consisting of skin barrier dysfunction, but also with immunological dysregulation and skin microbiota changes having a central role. In humans, establishment of the gut microbiota in early life influences the development of allergies, among others also atopic dermatitis in children. To the author’s knowledge, there is currently no study comparing the gut microbiome between allergic and healthy dogs. We now present results demonstrating that allergic dogs have a significantly different gut microbiota when compared to healthy control dogs. Further investigations including a larger number of dogs are now required to confirm these results, in addition to studies utilizing novel interventions targeting the gut microbiota.

**Abstract:**

Human studies show that in addition to skin barrier and immune cell dysfunction, both the cutaneous and the gut microbiota can influence the pathogenesis of atopic diseases. There is currently no data on the gut-skin axis in allergic canines. Therefore, the aim of this study was to assess the bacterial diversity and composition of the gut microbiome in dogs with atopic dermatitis (AD). Stool samples from adult beagle dogs (n = 3) with spontaneous AD and a healthy control group (n = 4) were collected at Days 0 and 30. After the first sampling, allergic dogs were orally dosed on a daily basis with oclacitinib for 30 days, and then re-sampled. Sequencing of the V3–V4 region of the 16S rRNA gene was performed on the Illumina MiSeq platform and the data were analyzed using QIIME2. The atopic dogs had a significantly lower gut microbiota alpha-diversity than healthy dogs (*p* = 0.033). In healthy dogs, a higher abundance of the families Lachnospiraceae (*p* = 0.0006), Anaerovoracaceae (*p* = 0.006) and Oscillospiraceae (*p* = 0.021) and genera *Lachnospira* (*p* = 0.022), *Ruminococcus*
*torques* group (*p* = 0.0001), *Fusobacterium* (*p* = 0.022) and *Fecalibacterium* (*p* = 0.045) was seen, when compared to allergic dogs. The abundance of *Conchiformibius* (*p* = 0.01), *Catenibacterium* spp. (*p* = 0.007), *Ruminococcus gnavus* group (*p* = 0.0574) and *Megamonas* (*p* = 0.0102) were higher in allergic dogs. The differences in alpha-diversity and on the compositional level remained the same after 1 month, adding to the robustness of the data. Additionally, we could also show that a 4-week treatment course with oclacitinib was not associated with changes in the gut microbiota diversity and composition in atopic dogs. This study suggests that alterations in the gut microbiota diversity and composition may be associated with canine AD. Large-scale studies preferably associated to a multi-omics approach and interventions targeting the gut microbiota are needed to confirm these results.

## 1. Introduction

Atopic dermatitis is a multifactorial disease, affecting millions of people and dogs worldwide. In the last decade, human studies showed that in addition to skin barrier and immune cell dysfunction, both the cutaneous and the gut microbiota are additional strategic players in AD pathogenesis [1]. The gut microbiota influences the allergen tolerance in many different ways, either directly interacting with T regulatory (Treg) cells or through the production of short-chain fatty acids (SCFA), post-translational host protein modification and activities on the epigenetic level [2,3]. Many studies showed that the pathogenesis of canine AD is very similar to the human counterpart, and therefore canine AD models represent an invaluable tool for human AD research [4]. To date, all studies in allergic dogs have focused only on the skin microbiota. Studies evaluating diversity and composition of gut microbiota in canine AD are lacking. Atopic dogs exhibit a significant dysbiosis with *Staphylococcus* as the predominant species in the flared skin [5], but these skin microbiota changes are rather considered to be a consequence than the cause of the disease [6]. The skin *Lachnospiraceae* frequency of puppies was found to be positively correlated with that of the Treg cells [7]. Moreover, a birth cohort study exploring the early life “window of opportunity”, identified gastrointestinal disturbances as one of the significant physiological foundations in the development of canine AD [8].

Therefore, the main objective of this work was to evaluate the diversity and composition of the gut microbiota in dogs with AD and compare these findings with those of healthy controls.

## 2. Materials and Methods

Stool samples from 3 adult beagle dogs (two male both 12 years old and one female 7 years old) with spontaneous AD and a control group consisting of 4 healthy dogs (three male with an average of 7 years and one female 9 years old) were collected at 2 time points (day 0 and 30) directly into PERFORMAbiome-GUT tubes (DNA Genotek, Ottawa, ON, Canada). The sample were immediately stored at −80 °C until processing. In addition to studying the time effect (one month sampling interval), we wanted to evaluate whether oclacitinib, a Janus kinase inhibitor (Apoquel, Zoetis, Delémont, Switzerland) is associated with gut microbiota changes. Therefore, after the initial sampling, the allergic dogs received orally once daily oclacitinib until the next re-sampling. The atopic dogs were allergic since years and received AD symptomatic treatment, which consisted of oclacitinib and occasionally topical glucocorticoids (Cortavance, Virbac, Carros, France). For this study, the atopic dogs did not receive any anti-inflammatory/anti-itch or antibiotic treatment for 1 month and several years before the first sampling, respectively. Moreover, the healthy dogs did not receive any drugs (including antibiotics) for several years. Both the allergic and heathy dogs were regularly de-wormed and vaccinated. All dogs are owned by the Vetsuisse Faculty University of Zurich. They live in the same environment (direct contact was possible) and are fed the same food. 

Extraction, lysis and DNA isolation were performed on stool samples, according to the manufacturer’s recommendation (Fast DNA Stool Mini Kit, Qiagen, Basel, Switzerland). Bead beating was run on a fastprep24 instrument (MP Biomedicals, Eschwege, Germany; 4 cycles of 45 s at speed 4) in 2 mL screwcap tubes containing 0.6 g of 0.1 mm glass beads. The concentration of the isolated DNA was assessed with PicoGreen measurement (Quant-iT™ PicoGreen™ dsDNA Assay Kit, Thermo Fisher Scientific, Basel, Switzerland), and integrity was checked for a random sample by agarose gel electrophoresis. The library preparation included sample quality control and Nextera two-step PCR amplification using primer set 515F × 806R (V3–V4 region of 16S rRNA), PCR product purification, quantification and equimolar pooling. The amplicon libraries paired-end sequencing (2 × 250 bp) was performed on an Illumina MiSeq platform at Microsynth AG (Balgach, Switzerland). 

The data was analyzed using QIIME2 (version 2020.2; https://docs.qiime2.org, access on 10 September 2022). The data was then denoised with DADA2 to merge the paired reads generating amplicon sequence variants (ASVs) to ensure a sufficient depth to capture most features [9]. An even sampling depth that was selected after performing alpha rarefaction module (60,000 reads per sample) was used to assess alpha- and beta-diversity measures. Alpha-diversity was assessed by Shannon and Faith’s phylogenetic diversity indexes. Beta-diversity was assessed using Bray Curtis and Jaccard distance measures and presented using principal coordinates analysis (PCoA) plots. Taxonomy was assigned to ASVs using the classify-learn Naïve Bayes classifier against the pre-trained Naïve Bayes silva-132-99-nb-classifier trained against Silva (release 132) full-length sequences [10]. The Wilcoxon test was used to assess the microbiota compositional differences between the AD and control groups at the different taxonomic levels, and the Permanova test was used for the beta analysis comparison. R (version 4.1.0; https://www.r-project.org, access on 10 September 2022) was used to perform all the statistical analysis and data visualization.

## 3. Results

### 3.1. Comparison of Bacterial Diversity Parameters between Allergic and Healthy Dogs

The reads obtained from the stool microbiota were analyzed, and a plateau of species richness (up to 200 ASVs) was achieved at 60,000 reads per sample to capture the diversity (Figure 1a). A clear difference in gut microbiota alpha diversity between allergic and healthy dogs was identified by Shannon alpha diversity index (*p* = 0.033) not only at day 0 but also 30 days later (Figure 1b). The Faith’s phylogenetic diversity index did not differ significantly at any time point (Figure 1c). Principal coordinate analysis by Bray-Curtis and Jaccard showed clustering of the bacterial gut microbiota of allergic against healthy dogs, reaching significance (*p* = 0.05) (Figure 1d,e).

### 3.2. Association of Gut Microbiota Composition with Atopic and Healthy States

Taxonomic classification showed ten phyla (Figure 2). Among the bacteria, the phylum Firmicutes was the most predominant in both groups (43%). Bacteroidetes and Proteobacteria contributed to 50% of other phyla.

Although no significant differences at phylum level between healthy and allergic dogs were seen, we identified 3 families enriched in healthy dogs: gram positive Anaerovoracaceae (*p* = 0.006), Ruminococcaceae (*p* = 0.024) and Peptostreptococcaceae (*p* = 0.0289), all belonging to phylum Firmicutes. At the genus level, a higher abundance of gram positive *Lachnospira* (Firmicutes, family Lachnospiraceae) (*p* = 0.0229), *Ruminococcus torques group* (Firmicutes, family Lachnospiraceae) (*p* = 0.0001), *Faecalibacterium* (Firmicutes, family Ruminococcaceae) (*p* = 0.0456), *UCG 005* (Firmicutes, family Oscillospiraceae) (*p* = 0.0523), *Peptoclostridium* (Firmicutes, family Peptostreptococcaceae) (*p* = 0.0289); and gram negative *Sutterella* (phylum Proteobacteria, family Sutterellaceae) (*p* = 0.0371) and *Fusobacterium* (phylum *Fusobacteriota*, family Fusobacteriaceae) (*p* = 0.022) was observed in healthy dogs compared to allergic dogs.

In contrast, allergic dogs exhibited a higher abundance of the genera *Conchiformibius* (phylum Proteobacteria, family Neisseriaceae) (*p* = 0.01), *Catenibacterium* (phylum Firmicutes, family Coprobacillaceae) (*p* = 0.0077), *Ruminococcus gnavus group* (phylum Firmicutes, family Lachnospiraceae,) (*p* = 0.0574) and *Megamonas* (phylum Bacteroidetes, family Veillonellaceae) (*p* = 0.0102). All genera which were significantly different between allergic and healthy dogs are shown in Figure 3.

The microbiota composition within the healthy and treated group did not change significantly neither by time nor by the treatment with the Janus kinase inhibitor oclacitinib, respectively, (healthy dogs: *p* = 0.39, atopic dogs: *p* = 0.51).

## 4. Discussion

To our knowledge this is the first study applying high throughput sequencing to analyze the composition of gut microbiota in dogs with AD. The most abundant phyla in canine gastrointestinal tract fall into five phyla: Firmicutes, Fusobacteria, Bacteroidetes, Proteobacteria, and Actinobacteria and were also identified in this study [11].

One of the main study findings is the significantly dysbiotic gut microbiota based on the lower alpha-diversity of dogs with spontaneous AD, which is in line with previous human studies [12,13]. Contrasting data in a minority of studies exist, but these studies were either of low-power or were only culture based [14,15,16]. A diverse microbial colonization during the critical time window in the first months of life is essential in order to limit default immune pathways and also negatively correlates with disease severity in already established disease [17]. 

Additionally, the gut dysbiosis was driven by compositional changes. Atopic dogs were associated with significantly higher prevalence of 4 genera: *Conchiformibius*, *Catenibacterium*, *Ruminococcus gnavus* group and *Megamonas.* This is the first report on the presence of *Conchiformibius* (family Neisseraceae) in the canine feces; furthermore, it was not found in other body sites. There are currently no reports on its role in human AD. Interestingly, a probiotic topical preparation containing *Vitreoscilla filiformis* from the same family, was reported to improve clinical signs of human AD [18]. Bacteria from the genus *Catenibacterium* (family Coprobacillaceae) were already isolated from canine faces especially in association with weight loss [19], but there are currently no reports on its role in allergic diseases. *Ruminococcus gnavus* group, consisting of mucolytic bacteria, was up to date not associated with allergic states, but was positively associated with parvovirus gastrointestinal infections in dogs and Crohn’s disease in humans [20,21], pointing toward a barrier-disrupting potential. *Megamonas*, a member of the family Veillonellaceae, was shown to have the highest sequence percentage in dogs consuming the prebiotic fructooligosacharide (FOS) inulin [22] and in humans it was found to have a lower relative abundance in human allergic rhinitis patients compared to healthy controls [23]. Based on these contradicting results, the role of different *Megamonas* species in diseases impacted by the gut microbiota needs further investigation.

Although the allergic dogs in this study did not have reduced abundance of *Bifidobacterium* and *Lactobacillus*, which are widely recognized protective microbial factors for atopic disease in humans [1], several other gut microbiota signature aberrations, related to immune homeostasis and barrier integrity, were observed. The allergic dogs showed a significantly reduced abundance of two members of the Lachnospiraceae family, *Lachnospira* and *Ruminococcus torques* group, which is in line with previous findings in atopic humans [24]. The Lachnospiracea family seems to be involved in gut epithelial barrier integrity and immune regulation, mainly through the production of SCFAs and mucin degradation [20]. They had also a significantly lower abundance of Ruminococcaceae family, an important source for SCFA, and which is also shaping the innate immunity through the toll-like receptor 2 and 4 mediated production of TNFa [25]. This family was also negatively associated with atopy in humans [26]. Additionally, a positive association between the reduced Ruminococcaceae house dust levels and the allergy incidence were recently reported [27]. Moreover, a lower abundance of the genus *Faecalibacterium,* an important representative of the Ruminococcaceae family was observed herein, corroborating data from asthmatic and inflammatory bowel disease (IBD) patients in humans [28], although contrasting reports also exist [29]. *Faecalibacterium prausnitzii*, has been considered beneficial to gut health through its production of SCFAs, in particular butyrate, which has anti-inflammatory effects and serves as the main energy source for the colonocytes, as reported in human and veterinary medicine [11,30]. Allergic dogs were literally depleted of bacteria belonging to the family Anaerovoracaceae; unfortunately, on the genus and species level no significant representative(s) could be identified. Similar findings were observed for the Peptostreptococcaceae family, and its genus *Peptoclostridium*, also their function in health and disease is currently unclear.

Additionally, *UCG 005* (family Oscillospiraceae), a member of the phylum Firmicutes, was significantly reduced in allergic dogs when compared to healthy controls. *UCG 005* was previously not reported in canine microbiome studies but was more abundant in allergic human patients [29]. Genus *Oscillospira* from the same family, correlates highly with SCFA production in piglets [31]. Peptostreptococcaceae family and the genus *Peptoclostridium*, which was just recently identified in canine faeces [19], were significantly reduced in allergic dogs. To date no reports on their significance in atopic disease exist.

The microaerophilic gram-negative *Sutterella* (phylum Proteobacteria), an already recognized taxa from canine feces, was less prevalent in allergic dogs feces, corroborating previous findings in allergic humans [12]. Contrasting reports on Sutterella relative abundances in allergic children exist and their function still needs to be elucidated [29]. Furthermore, allergic dogs also had a lower abundance for *Fusobacterium* sp. from the phylum Fusobacteriota. This is in contrast with data from allergic humans, indicating that *Fusobacterium* is playing a different role in the GI tract of dogs, as recently reported [11]. 

That the gut microbiota is an important player in the disease pathogenesis was recently shown by interventional studies using fecal microbial transplantation (FMT) in patients with AD. These studies showed that FMT is safe and efficacious in controlling clinical signs of AD in dogs and humans [32,33]. Therefore, there is a high need for future studies in this field.

We also showed that a 30-day course of oral treatment with a Janus kinase inhibitor oclacitinib was not associated with changes in the gut microbiota. This finding is in line with a previous study in mice treated with baricitinib, a small molecule targeting the Janus kinase pathway [34] and with a study using oral prednisolone in dogs [35]. This data would be useful for future interventional studies targeting the gut microbiota in atopic dog and to understand better the potential of confounding factors (non-antiobiotic drugs) on the gut microbiota outcomes.

One weakness of our current report is its low power, as only 3 allergic and 4 healthy dogs were included. This was counteracted by controlling for most influential gut microbiota environmental (food, lifestyle and living environment) and genetic factors (all dogs were Beagles), as shown in previous human studies [36,37]. In dogs, there is currently a large body of evidence that the diet exerts a significant impact on the gut microbiota in this species [38,39], but there is no data on the genetic influence. This combined with the fact that the breed and the environment were controlled, gives our report a crucial edge, since both factors have been shown to be important in the pathogenesis of AD [40,41]. We hypothesize that controlling for these confounding factors enabled the recognition of statistically significant differences between the allergic and healthy group. The fact that the differences between these groups remained significant after 1 month adds to the robustness of the data described herein. Whether the differences in the described levels of different bacteria is a causative, resultant or only accompanying factor for canine AD remains to be elucidated in high-power studies. Additionally, further transcriptome-based studies should evaluate the functional aspects of the recognized potential microbial targets.

## 5. Conclusions

The reduced diversity and differences of specific bacterial taxa suggest that the gut microbiota may play a role in the pathogenesis of canine AD. The herein identified potentially beneficial and pathogenic bacterial targets should be further explored in large-scale studies as possible disease or therapy monitoring biomarkers. Furthermore, future interventional studies encompassing multi-omics approaches represent the next steps toward unravelling the presence and pathophysiology of the gut-skin axis in this species.

## Figures and Tables

**Figure 1 animals-12-02377-f001:**
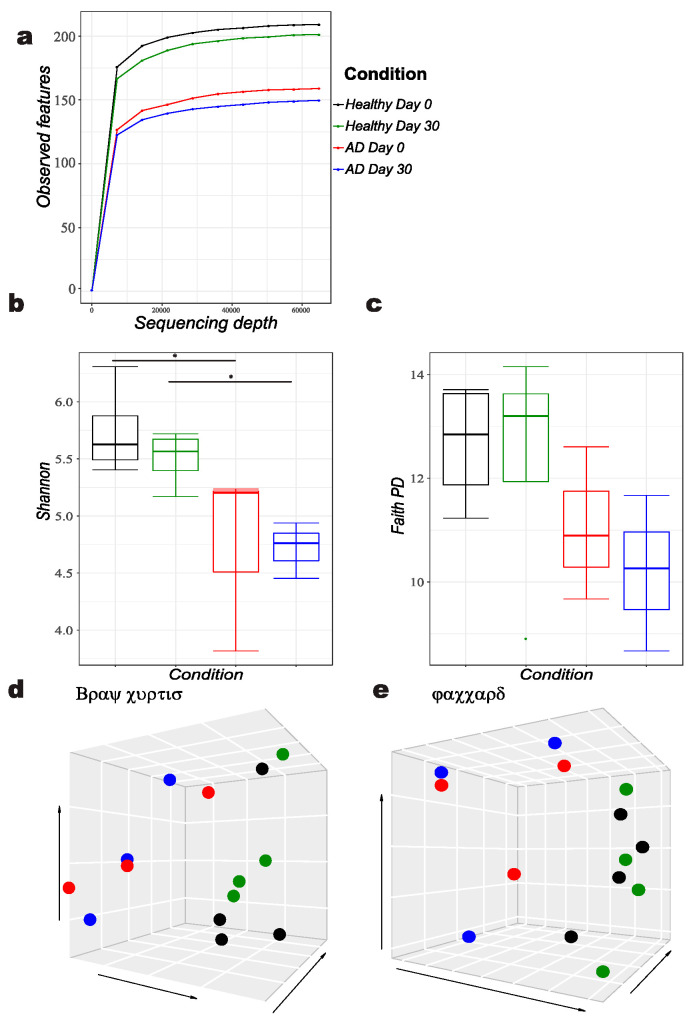
Diversity analysis of atopic versus healthy dogs, showing following features: (**a**) Alpha rarefaction, (**b**) Shannon alpha diversity index B, (**c**) Faith’s phylogenetic diversity (PD) index, Principal coordinates analysis (**d**) Bray Curtis and (**e**) Jaccard diversity matrices.

**Figure 2 animals-12-02377-f002:**
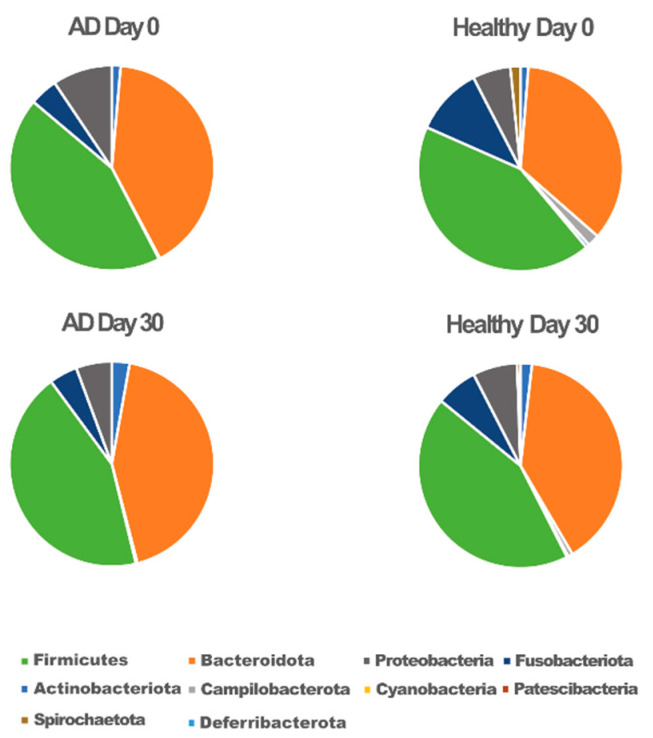
Microbiota composition at the phylum level for atopic (AD) and healthy dogs.

**Figure 3 animals-12-02377-f003:**
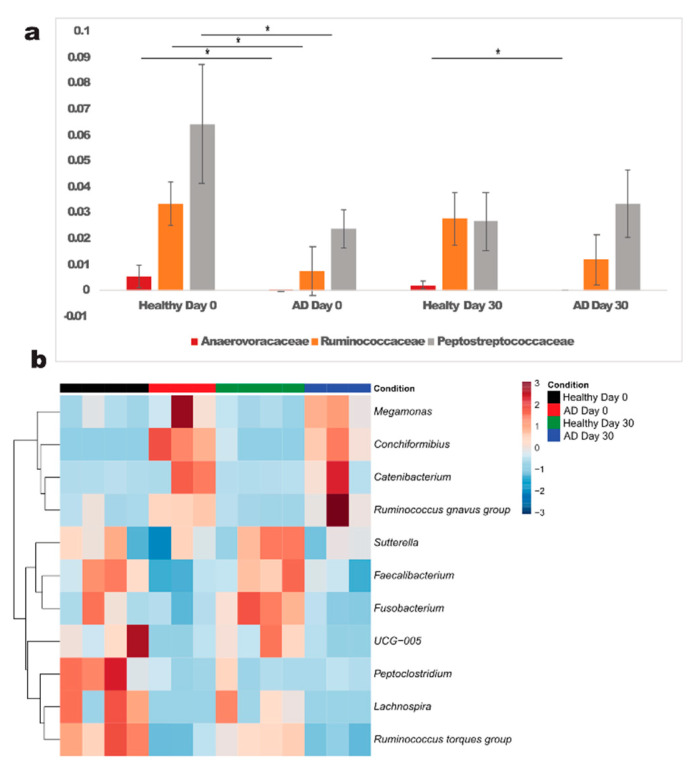
Significantly different families (**a**) and genera (**b**) in atopic and healthy dogs at day 0. *—significant differences were detected between healthy and atopic dogs (*p* < 0.05); AD—atopic dogs.

## Data Availability

Not applicable.

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
