# Peer review of "Comparison of the Gut Microbiome between Atopic and Healthy Dogs—Preliminary Data"

_animals, 2022, doi:10.3390/ani12182377_

Round 1
Reviewer 1 Report
This is the personal review of the article entitled “Comparison of the gut microbiome between atopic and healthy dogs” from Rostaher et al.
The authors reported the comparison of the gut microbiome composition between healthy dogs and dogs with atopic dermatitis (AD), claiming that this is the first report regarding this type of comparison.
Overall the study is interesting but in my opinion, it lacked some information and scientific consistency. Here there are my comments.
Major comments:
Stool samples from 3 adult beagle dogs with spontaneous AD and a control group consisting of 4 healthy dogs: I am afraid that this number of dogs is too limited to do any type of comparisons. Moreover, the dogs are more in a senior phase rather than an adult phase.
After you mentioned that you used qiime2 you should cite the authors: Bolyen E, Rideout JR, Dillon MR, Bokulich NA, Abnet CC, Al-Ghalith GA, Alexander H, Alm EJ, Arumugam M, Asnicar F, Bai Y, Bisanz JE, Bittinger K, Brejnrod A, Brislawn CJ, Brown CT, Callahan BJ, Caraballo-Rodríguez AM, Chase J, Cope EK, Da Silva R, Diener C, Dorrestein PC, Douglas GM, Durall DM, Duvallet C, Edwardson CF, Ernst M, Estaki M, Fouquier J, Gauglitz JM, Gibbons SM, Gibson DL, Gonzalez A, Gorlick K, Guo J, Hillmann B, Holmes S, Holste H, Huttenhower C, Huttley GA, Janssen S, Jarmusch AK, Jiang L, Kaehler BD, Kang KB, Keefe CR, Keim P, Kelley ST, Knights D, Koester I, Kosciolek T, Kreps J, Langille MGI, Lee J, Ley R, Liu YX, Loftfield E, Lozupone C, Maher M, Marotz C, Martin BD, McDonald D, McIver LJ, Melnik AV, Metcalf JL, Morgan SC, Morton JT, Naimey AT, Navas-Molina JA, Nothias LF, Orchanian SB, Pearson T, Peoples SL, Petras D, Preuss ML, Pruesse E, Rasmussen LB, Rivers A, Robeson MS, Rosenthal P, Segata N, Shaffer M, Shiffer A, Sinha R, Song SJ, Spear JR, Swafford AD, Thompson LR, Torres PJ, Trinh P, Tripathi A, Turnbaugh PJ, Ul-Hasan S, van der Hooft JJJ, Vargas F, Vázquez-Baeza Y, Vogtmann E, von Hippel M, Walters W, Wan Y, Wang M, Warren J, Weber KC, Williamson CHD, Willis AD, Xu ZZ, Zaneveld JR, Zhang Y, Zhu Q, Knight R, and Caporaso JG. 2019. Reproducible, interactive, scalable and extensible microbiome data science using QIIME 2. Nature Biotechnology 37: 852–857. https://doi.org/10.1038/s41587-019-0209-9
Which version of qiime2 did you use?
You said that you used the silva database (release 132) for the taxonomic annotation: can you explain why you used the 132 release instead of the new one? (release 138 or 138.1).
You mentioned that there are “contrasting data in a minority of studies” and you cited a review. It’s always good to cite the original studies as well, in order to give the authors credit for what they did.
My biggest concern regards what you observed in the gut microbiome of AD dogs: they had a lower alpha diversity compared to the healthy dogs, and indeed you observed a reduction, in terms of abundances, of taxa that are related to the immune homeostasis and barrier integrity. The point is that the AD dogs were treated, from the beginning of the study until the end, with oclacitinib, and you do not have a group of AD dogs that were not under treatment. How can you be sure that the lower alpha diversity and the lower abundances of certain bacteria are not an effect of the treatment?
Minor comments:
1) There are differences in terms of font characters all over the manuscript. Try to uniforms.
2) There are differences in italics and lowercase block letters on the bacteria all over the manuscript. Genus and species level should be in italic; from family to phylum level can be in lowercase block letters.
3) There are several sentences that are not very clear, together with some typos. Try to read the manuscript carefully.
Abstract:
sequencing of the v34 region; this is a refuse.
Lower gut microbiota diversity: alpha or beta? Specified
Introduction:
T regulatory cells (Treg): the abbreviation should be T-reg.
Materials and methods:
BID and SID: you should explain what those terms mean before writing the abbreviation.
Results:
The figures are way too small and not with a high-quality resolution. Dots, corresponding to subjects, in the PCoAs are too small and It is not easy to distinguish the colors. Moreover, it is hard to see a clusterization with just 7 subjects in total.
Discussion:
You wrote that there is no report of Megamonas in the dogs and human field, but I am sure that this is not correct.
Author Response
Dear editor, dear reviewers
Thank you so much for considering our manuscript for publication. We found your input very constructive and helpful. We hope that the quality of our manuscript after the revision is much better.
As our data base on a very small population, we can only talk about preliminary results. Therefore, we changed the title to: Comparison of the gut microbiome between atopic and healthy dogs – preliminary data.
We addressed the materials and methods section, which was not clear about the drug history of the dogs and that we wanted to test the effect of the Janus kinase inhibitor oclacitinib. Additionally, we corrected all typos regarding the nomenclatures associated with the gut microbiota (spelling of the Phila, genera, families; V3-V4 region…). Our biostatistician also addressed the questions regarding the data analysis and improved the PCoA Figures in Figure 1, he also improved the overall resolution of the figures.
Furthermore, we also extended the discussion and added the following
- more data for Megamonas
- data on confounding factors for gut microbiome (and atopic dermatitis), strengthening our rationale to publish study findings with this low power
- that we add a small piece regarding the influence of oclacitinib on the gut microbiota, it was already previously shown that Prednisolone is not affecting the gut microbiota in dogs and in mice the same was observed with the same class drug as oclacitinib (baricitinib)
- information that there are recently also published studies on the efficacy and safety of faecal microbial transplantation in not only dogs but also humans with atopic dermatitis. Based on this, we feel that there is a strong rationale to perform studies in this field.
- With the above mentioned data we also automatically extended the reference list from 27 to 42. As these changes we could not show by the tracked changes word feature, we highlighted them by red colour font, hoping that this is not causing problems.
Prof. Chris Asquith (see acknowledgements), a native English speaking scientist reviewed and corrected our manuscript.
Our wish is to prove our findings from this preliminary report in an upcoming large-scale multi-omics study to explain better the role of the gut microbiota in atopic dogs. This publication would in addition to the existing literature make a good basis to follow up this “story”.
Thank you for your consideration of our manuscript.
Looking forward to hearing from you.
Ana Rostaher
REVIEWER 1
Overall the study is interesting but in my opinion, it lacked some information and scientific consistency. Here there are my comments.
Major comments:
- Stool samples from 3 adult beagle dogs with spontaneous AD and a control group consisting of 4 healthy dogs: I am afraid that this number of dogs is too limited to do any type of comparisons. Moreover, the dogs are more in a senior phase rather than an adult phase.
Answer:
We completely agree with you. Therefore, we pointed out that this is the major limitation of this study in the last part of the discussion. It was just a “lucky” situation to have the chance to identify simultaneously 3 dogs with spontaneous AD in a controlled environment. Of course, it would be better if there would be more allergic dogs and if they would be younger at the time of sampling. But at the earlier time we did not envisage the gut microbiota as an important pathogenetic player. Our previous studies focused first on canine AD clinical criteria, Treg cells, environmental factors of AD and then also skin microbiota. With the skin microbiota study in a birth cohort, we could show that skin microbiota changes are not one of the primary drivers in AD pathogenesis. We agree with you, these results can be regarded only as preliminary, so we changed the title accordingly. If the 3 atopic dogs in the current study would live in different environments and eating different food and not having the same genetic background, we would be sceptical to publish this data. But as the dogs were controlled for the most important confounding factors for the microbiota, we believe that this preliminary data can be shared with the scientific community. We plan a large-scale multi-omics study for the future and hope we can confirm (or refute) and better characterize/explain the findings from this preliminary study. Especially we believe that probably there will be a subset of atopic dogs, having gut microbiota changes as important pathophysiological factor, which we will be able to identify by multi-omics approach. With this, hopefully, in the future new (efficient) microbiota targeting therapeutic approaches will arise.
Regarding the second part of your commentquestion. As the disease severity remained stable by time, we expect that the major patho-physiological characteristics were not affected by aging.
- After you mentioned that you used qiime2 you should cite the authors: https://doi.org/10.1038/s41587-019-0209-9
Answer: We added this reference.
- Which version of qiime2 did you use?
Answer: Qiime version 2020.2 was used. We added the information
- You said that you used the silva database (release 132) for the taxonomic annotation: can you explain why you used the 132 releases instead of the new one? (Release 138 or 138.1).
Answer: Starting from the release of 138 they switched the way of performing clustering and at the time of analysis we were more confident about the USEARCH algorithm to perform the clustering. As we are already working on additional studies using already the silva database release 132, we want to uniform the analysis method so we can compare the results.
- You mentioned that there are “contrasting data in a minority of studies” and you cited a review. It’s always good to cite the original studies as well, in order to give the authors credit for what they did.
Answer: We totally agree and added the references of the single original studies (Ref 14-16).
- My biggest concern regards what you observed in the gut microbiome of AD dogs: they had a lower alpha diversity compared to the healthy dogs, and indeed you observed a reduction, in terms of abundances, of taxa that are related to the immune homeostasis and barrier integrity. The point is that the AD dogs were treated, from the beginning of the study until the end, with oclacitinib, and you do not have a group of AD dogs that were not under treatment. How can you be sure that the lower alpha diversity and the lower abundances of certain bacteria are not an effect of the treatment?
Answer: thank you for picking up on this one. We did not explain well the materials & methods, the initial part was too short, and we took for granted that everything is clear. Reading this now after few weeks, we agree it is not clear.
First, here is an explanation for you: We sampled the healthy and atopic dogs before treatment (1 month wash-out period for oclacitinib) to evaluate difference for the AD status (goal 1). Then we re-sampled the dogs after 30 days (AD dogs received in this time 30 days oclacitinib). With this we wanted to show if there is any effect of oclacitinib on the gut microbiota (goals 2) as in the future we would like to use oclacitinib in interventional study with fecal transplantation for symptomatic relieve, and this would be of large importance to plan this study. Most of the interventional studies in atopic dogs have to allow the dogs symptomatic treatment, we already have data for the prednisolone, which does not affect significantly the gut microbiota (https://pubmed.ncbi.nlm.nih.gov/25229475) and now we wanted have at least preliminary data also for oclacitinib.
Additionally, by analyzing the gut microbiome horizontally on two occasions, we wanted to strengthen the results for the first part. That means, if we could show that atopic and healthy dogs keep being different in diversity and compositional markers longitudinally, this would add significantly to the data robustness.
Now we changed the text and hope it makes more sense:
“Stool samples from 3 adult beagle dogs (two male both 12 years old and one female 7 years old) with spontaneous AD and a control group consisting of 4 healthy dogs (three male with an average of 7 years and one female 9 years old) were collected at 2 time points (day 0 and 30) directly into PERFORMAbiome-GUT tubes (DNA Genotek, Ottawa, Canada). The sample were immediately stored at -80’C until processing. In addition to studying the time effect (one month sampling interval), we wanted to evaluate if oclacitinib, a Janus kinase inhibitor (Apoquel, Zoetis, Germany) is associated with gut microbiota changes. Therefore, after the initial sampling, the allergic dogs received orally once daily oclacitinib until the next re-sampling. The atopic dogs were allergic since years and received AD symptomatic treatment, which consisted of oclacitinib and occasionally topical glucocorticoids (Cortavance, Virbac, France). For this study, the atopic dogs did not receive any anti-inflammatory/anti-itch or antibiotic treatment 1 month and several years before the first sampling, respectively. Moreover, the healthy dogs did not receive any drugs (including antibiotics) for several years. Both, the allergic and heathy dogs were regularly de-wormed and vaccinated. All dogs are owned by the Vetsuisse Faculty University of Zurich. They live in the same environment (direct contact was possible) and are fed the same food.”
- Minor comments:
- There are differences in terms of font characters all over the manuscript. Try to uniforms.
Answer: Thank you for the comment, we removed the typos.
2) There are differences in italics and lowercase block letters on the bacteria all over the manuscript. Genus and species level should be in italic; from family to phylum level can be in lowercase block letters.
Answer: Thank you for the comment, we consistently wrote the Phylum, Family, genus and species
3) There are several sentences that are not very clear, together with some typos. Try to read the manuscript carefully – Ask English speaking person
Answer: You are right, we asked an English native speaker to review the article, and hope the Grammar and style are now better.
- Abstract: sequencing of the v34 region; this is a refuse. Lower gut microbiota diversity: alpha or beta? Specified
Answer: we improved to V3-V4 region. We corrected that alpha diversity was affected
- Introduction: T regulatory cells (Treg): the abbreviation should be T-reg.
Answer: We wrote several articles on Treg cells with co-authorship with Prof. CA Akdis who is one of the top specialists in this filed in human allergology and he confirmed the Treg abbreviation, and the journals also accepted it. In addition, when we checked Pubmed for 2022 and 2021 for publications with T regulatory cells, the Treg abbreviation is widely used. I hope with this we can convince you to leave this nomenclature as it is.
- Materials and methods: BID and SID: you should explain what those terms mean before writing the abbreviation.
Answer: Thank you for this remark. We thought that BID/SID are well recognized abbreviations in scientific journals (so we did not describe it). As we do not repeat this terminology within this manuscript, we now added the following text: twice daily/once daily. - Results: The figures are way too small and not with a high-quality resolution. Dots, corresponding to subjects, in the PCoAs are too small and It is not easy to distinguish the colors. Moreover, it is hard to see a clusterization with just 7 subjects in total.
Answer: We addressed this and provide new figures.
- Discussion:
You wrote that there is no report of Megamonas in the dogs and human field, but I am sure that this is not correct.
Answer: Thank you for your comment, which was very helpful. You are right we did find one article in humans (Zhu et al 2020: Gut microbial characteristics of adult patients with allergy rhinitis), which reported of a lower relative abundance of Megamonas in allergic rhinitis patients compared to healthy controls. We included this article and it’s findings in our manuscript. Additionally, we also found an article in dogs which showed that Megamonas was more abundant after feeding Inulin, written by Beloshapka et al 2013 (Fecal microbial communities of healthy adult dogs fed raw meat-based diets with or without inulin or yeast cell wall extracts as assessed by 454 pyrosequencing), reference 22.

Reviewer 2 Report
Very interesting study but preliminary results (too low numbers of observations)

Author Response
Dear editor, dear reviewers
Thank you so much for considering our manuscript for publication. We found your input very constructive and helpful. We hope that the quality of our manuscript after the revision is much better.
As our data base on a very small population, we can only talk about preliminary results. Therefore, we changed the title to: Comparison of the gut microbiome between atopic and healthy dogs – preliminary data.
We addressed the materials and methods section, which was not clear about the drug history of the dogs and that we wanted to test the effect of the Janus kinase inhibitor oclacitinib. Additionally, we corrected all typos regarding the nomenclatures associated with the gut microbiota (spelling of the Phila, genera, families; V3-V4 region…). Our biostatistician also addressed the questions regarding the data analysis and improved the PCoA Figures in Figure 1, he also improved the overall resolution of the figures.
Furthermore, we also extended the discussion and added the following
- more data for Megamonas
- data on confounding factors for gut microbiome (and atopic dermatitis), strengthening our rationale to publish study findings with this low power
- that we add a small piece regarding the influence of oclacitinib on the gut microbiota, it was already previously shown that Prednisolone is not affecting the gut microbiota in dogs and in mice the same was observed with the same class drug as oclacitinib (baricitinib)
- information that there are recently also published studies on the efficacy and safety of faecal microbial transplantation in not only dogs but also humans with atopic dermatitis. Based on this, we feel that there is a strong rationale to perform studies in this field.
- With the above mentioned data we also automatically extended the reference list from 27 to 42. As these changes we could not show by the tracked changes word feature, we highlighted them by red colour font, hoping that this is not causing problems.
Prof. Chris Asquith (see acknowledgements), a native English speaking scientist reviewed and corrected our manuscript.
Our wish is to prove our findings from this preliminary report in an upcoming large-scale multi-omics study to explain better the role of the gut microbiota in atopic dogs. This publication would in addition to the existing literature make a good basis to follow up this “story”.
Thank you for your consideration of our manuscript.
Looking forward to hearing from you.
Ana Rostaher
REVIEWER 2
- Abstract :
Sequencing of the V34 region Sequencing of the V3-V4 region of the 16S rRNA gene .
Answer: We corrected this to V3-V4 region of the 16S rRNA gene.
Materials and Methods
- ? Dogs used for the observations were « laboratory » dogs (mentioned latter) , the healthy dogs were also Beagles ? (not clear) and the 7 dogs lived in the same environment …with direct contacts ? (not clear)
Answer: you are right we wanted to write this paragraph short and took for granted that everything is clear. Reading now it is really confusing. We sampled the healthy and atopic dogs before treatment to evaluate the effect on oclacitinib on the gut microbiota. The AD dogs did not receive any symptomatic treatment for allergies at least 1 month before study begin.
Now we changed the text and hope it is clearer:
“Stool samples from 3 adult beagle dogs (two male both 12 years old and one female 7 years old) with spontaneous AD and a control group consisting of 4 healthy dogs (three male with an average of 7 years and one female 9 years old) were collected at 2 time points (day 0 and 30) directly into PERFORMAbiome-GUT tubes (DNA Genotek, Ottawa, Canada). The sample were immediately stored at -80’C until processing. In addition to studying the time effect (one month sampling interval), we wanted to evaluate if oclacitinib, a Janus kinase inhibitor (Apoquel, Zoetis, Germany) is associated with gut microbiota changes. Therefore, after the initial sampling, the allergic dogs received orally once daily oclacitinib until the next re-sampling. The atopic dogs were allergic since years and received AD symptomatic treatment, which consisted of oclacitinib and occasionally topical glucocorticoids (Cortavance, Virbac, France). For this study, the atopic dogs did not receive any anti-inflammatory/anti-itch or antibiotic treatment 1 month and several years before the first sampling, respectively. Moreover, the healthy dogs did not receive any drugs (including antibiotics) for several years. Both, the allergic and heathy dogs were regularly de-wormed and vaccinated. All dogs are owned by the Vetsuisse Faculty University of Zurich. They live in the same environment (direct contact was possible) and are fed the same food.”
- Statistics on groups with n=3 or n=4 are really a challenge and a big source of errors. This has been mentioned in the discussion but a greater caution should be addressed in interpreting the results; it is a preliminary study.
Answer: You are completely correct. This was a preliminary study, but a larger study will follow (if we receive the grant, crossing fingers). We feel that it is important share data from this preliminary study with the scientific community. We show that there is a possibility that the gut microbiome is involved in the canine AD pathogenesis, giving the direction to further studies exploring this on large scale. We changed the title to: Comparison of the gut microbiome between atopic and healthy dogs – preliminary data
- Results
A comparison of the biodiversity between the two groups (4 issues: D0 and D30) using a 3D representation using nonmetric dimensional scaling, based upon the Bray–Curtis dissimilarity matrix should be very interesting, giving a better overall view (beta-diversity)… something like that:
Answer: Thank you for this comment, we prepared new figures, which are now hopefully giving a better overview.
- Discussion (should be extended a bit)
Answer: We extended the discussion with following (last part pf the discussion):
“That the gut microbiota is an important player in the disease pathogenesis was recently shown by interventional studies using faecal microbial transplantation (FMT) in patients with AD. These studies showed that FMT is safe and efficacious in controlling clinical signs of AD in dogs and humans [1, 2]. Therefore, there is a high need for future studies in this field.
We also showed, that a 30-day course of oral treatment with a Janus kinase inhibitor oclacitinib, was not associated with changes in the gut microbiota. This finding is in line with a previous study in mice treated with baricitinib, a small molecule targeting the Janus kinase pathway [3] and with a study using oral prednisolone in dogs [4]. This data would be useful for future interventional studies targeting the gut microbiota in atopic dog and to understand better the potential of confounding factors (non-antiobiotic drugs) on the gut microbiota outcomes.
One weakness of our current report is its low power, as only 3 allergic and 4 healthy dogs were included. This was counteracted by controlling for most influential gut microbiota environmental (food, life-style and living environment) and genetic factors (all dogs were Beagles), as shown in previous human studies [5, 6]. In dogs, there is currently a large body of evidence that the diet is exerting significant impact on the gut microbiota in this species [7, 8], but there is no data on the genetic influence. This combined with the fact that we control for, the breed and the environment gives our report a crucial edge, since both factors have been shown to be important in the pathogenesis of AD [9, 10]. We hypothesize, that controlling for these confounding factors, enabled the recognition of statistically significant differences between the allergic and healthy group. The fact, that the differences between these groups remained significant after 1 month, adds to the robustness of the data described herein. Whether the differences in the described levels of different bacteria is a causative, resultant or only accompanying factor for canine AD remains to be elucidated in high-power studies. Additionally, further transcriptome- based studies should evaluated the functional aspects of the recognized potential microbial targets.”
Reading the discussion, I do not understand why oclacitinib was used. What was the goal? (not discussed) Did the authors expect a change in the gut microbiota with oclacitinib? Or was it because the atopic dogs couldn't stay without any treatment (what did they get in the years before? They were quite old (chronic) atopic dogs.
Answer: You are right we missed to explain this. The dogs also received oclacitinib or topical glucocorticoids occasionally before, but there were without oclacitinib medication for 1 month, before first sampling. We wanted to see if oclacitinib, which might be used in future interventional studies utilizing gut microbiota modification is a suitable symptomatic medication not affecting the gut microbiota.
We explain this now in the discussion:
We also showed that a 30-day course of oral treatment with a Janus kinase inhibitor oclacitinib, was not associated with changes in the gut microbiota. This finding is in line with a previous study in mice treated with baricitinib, a small molecule targeting the Janus kinase pathway [3] and with a study using oral prednisolone in dogs [4]. This data would be useful for future interventional studies targeting the gut microbiota in atopic dog and to understand better the potential of confounding factors (non-antiobiotic drugs) on the gut microbiota outcomes.
As said this is a very interesting preliminary study but it requires confirmation with a larger number of dogs tested and also an assessment of the stability of the microbiota with time.
Answer: we completely understand your point, this was a preliminary study for which we feel the data can be published, but we cannot draw strong conclusion, these results should be confirmed in a larger scale study. We mention this in the conclusion, abstract and after review also the title.

Round 2
Reviewer 1 Report
Dear authors,
Thank you for answering all my concerns and questions. I now found the article ready for the publication.
Best regards.